# Effect of Dynamic Loading Conditions on Maximizing Energy Dissipation of Metallic Dampers

**Ji Woon Park [1], Ji-Hoon Yoon [2], Gil-Ho Yoon [2] and Yun Mook Lim [1,*]**

[1] Department of Civil and Environmental Engineering, Yonsei University, 50 Yonsei-ro, Seodaemoon-gu, Seoul 03722, Korea; jwp1021@yonsei.ac.kr

[2] Department of Mechanical Engineering, Hanyang University, 222 Wangsimni-ro, Seongdong-gu, Seoul 04763, Korea; jihoon9212@gmail.com (J.-H.Y.); gilho.yoon@gmail.com (G.-H.Y.)

[*] Correspondence: yunmook@yonsei.ac.kr

**Abstract:** Diversification of the optimum designs is practical for metallic dampers due to their advantages of low cost, stability, and ease of fabrication. Therefore, this paper presents a novel approach—dynamic optimization—to derive various optimum shapes of metallic dampers that will dissipate the greatest amount of seismic energy. Specifically, this study proposes a conceptual metallic damper for bridges as a target model to investigate and develop the optimization method. First, an optimizing system was constructed by combining an optimization algorithm (sequential quadratic programming, SQP) with finite element analysis. In a conventional optimization process, energy dissipation capability and stiffness of the metallic damper increases under given static loadings. However, the conventional process fails to diversify the optimized shapes and results in less energy dissipated in conditions with relatively small ground motions due to the increased stiffness. Therefore, a novel method with a simple numerical model for dynamic optimization was devised with additional spring sets and concentrated masses. By utilizing this model, the optimized results under relatively high acceleration conditions were similar to the statically optimized cases, while the other cases showed different trends of optimum shapes. These unconventional results demonstrate decreased stiffness in static analysis, but eventually exhibit higher energy dissipation during small earthquakes.

**Keywords:** metallic dampers; shape optimization; energy dissipation; structural analysis; seismic design

## 1. Introduction

Beyond simply increasing the stiffness of structures for seismic reinforcements, modern designs withstand earthquakes by installing additional dampers. Among the various dampers, hysteretic dampers have relatively low cost, excellent durability, and high reliability because of their sustainable nature, as they use the plasticity of materials [1,2]. Hysteretic dampers, such as steel shear dampers or unbounded braces, dissipate seismic energy through the plastic deformation of materials, which are generally metals. The metallic dampers can be applied to retrofit buildings with a relatively small scale of deformation [3,4]. In addition, based on their ease of installation, they can potentially be applied to structures under various conditions, such as transmission lines or overhead contact systems [5,6]. Accordingly, many researchers have proposed novel concepts of metallic dampers using assorted approaches, such as a crawler steel damper [7], steel plates with a vertical free mechanism [8], a hysteretic damper with torsional yielding [9], rippled of curved metallic plate dampers [10,11], and a steel tube-in-tube damper [12]. Moreover, the efficiencies of existing metallic dampers have also been improved to facilitate installation in large-scale structures. For example, shape-optimization studies for a shear slip damper [13], a steel shear panel damper (SSPD) [14], a U-shaped damper [1], and a coupling beam metal damper [15] have been conducted. However, none of these studies developed or evaluated their dampers with respect to different capacity requirements under various installation

conditions. Furthermore, the conventional studies optimize dampers by increasing the energy dissipation capacity and stiffness, mostly depending on static experiments or analysis. Therefore, sometimes statically optimized dampers might not perform as expected under real earthquakes, which are dynamic, especially when the earthquakes are relatively small.

In this context, this study aims to develop a progressive development process for metallic dampers with a simple hypothetical installation concept. This process is based on the methodology of shape optimization under dynamic loading conditions. Most studies on the optimized shape of metallic dampers have focused on flexural yielding; that is, in-plane bending. In our study, with the same notion of flexural yielding, the simplest concept of a metallic damper is presented. However, unlike static optimization studies that have been conducted several times in the past, this study presents a new methodology of dynamic optimization of metallic dampers. Based on this concept, this study suggests various approaches to optimize the geometry of the damper.

As metallic dampers can be manufactured in various shapes, this study focuses on developing specialized shapes for different target capacities. Several diverse conditions were applied to establish the distinctive optimization methodology of this study in order to derive a variety of shapes optimized for different conditions. Through alteration of the optimization conditions, the effects of the maximum stroke size, the number of loading cycles, and the presence of additional control variables were studied. Consequently, a novel approach was constructed with the application of a simple model for the optimization process, including dynamic analysis. The overall development process was conducted via an optimization algorithm combined with the finite element (FE) software ABAQUS 6.14. This calculation process maximized the energy dissipation capability of the damper; as a result, the optimal shapes were derived.

## 2. Design Concept and Optimization Approaches

An ideal damper should be reliable in performance, dissipate sufficient energy, and exhibit stable hysteretic behavior against the seismic load. However, conventional studies primarily considered the static loading condition to optimize and evaluate metallic dampers. To address this problem by comparing two different approaches, static and dynamic, a simple prismatic metallic damper with a hypothetical installation concept is proposed. Based on this concept, a basic process of the analysis and optimization is presented in the following sections of this study.

### 2.1. Design Concept of a Prismatic Metallic Damper

As noted, an ideal damper should be reliable in performance, dissipate sufficient energy, and exhibit stable hysteretic behavior against the seismic load. The minimum qualification of reliability can be guaranteed with respect to metallic dampers, since they dissipate seismic energy through plastic deformation after yielding [16]. However, increasing the energy dissipation capacity and the stability is necessary for developing an effective metallic damper [15]. In terms of capacity and stability, the two key points can be summarized as follows. First, the energy dissipation should be maximized [17]. This maximization can be expressed as the increase in the area of the hysteresis loop of the metallic damper under cyclic loading. To achieve the efficiency, a total volume of the material usage should be constrained in the maximization process. Second, the shape or material components of the damper should not cause a stress concentration or excessive plastic strain at a particular point in the member [18]. Through these objectives, a better-performing metallic damper can be obtained.

To achieve the goals presented above, researchers can either improve existing shapes of metallic dampers, or find new optimal shapes, via one of two methods: (1) an experimental evaluation based on a geometry derivation through mechanical calculation, or (2) an optimization process using iterative computing. In the former case, the performance is evaluated via several experiments based on a few different shapes—usually 5–10 shapes [1,8,10,17,19–21]. On the other hand, the latter process combines numerical

analysis with a computational optimization scheme to find the optimal shape from repetitive computation. For such an optimization, various techniques, such as Tabu search [22], the simulated annealing algorithm [13,23], the response surface methodology [14], evolutionary optimization [24–26], and topology optimization [27–29], can be applied. All of these related studies are based on FE analysis; it has been proven that the analysis results of FE modeling (FEM) are very similar to the experimental results [1,8,10,13,14,22].

In general, computational optimization based on FEM is applied for upgrading metallic dampers already in use. In contrast, trial-and-error methods based on experimental development have been commonly utilized to develop new types of metallic dampers. However, in this paper, computational optimization was not exploited to improve an existing form, but to find a new and optimal metallic damper geometry. The design concept of this study directly focuses on investigating optimum shape derivations for metallic dampers with respect to different static or dynamic loading conditions.

In this study, a simple concept of a metallic damper is proposed—one that is designed to be additionally installed between the superstructure and the lower column of the bridge. The conceptual installation conditions are illustrated in Figure 1. The basic size and shape of the damper is presented in the form of a rectangular parallelepiped with a total length of 1200 mm, a width of 800 mm, and a thickness of 40 mm (Figure 2). Only one end of the damper is allowed to move in one lateral direction; the other perpendicular movements or moments are fixed. It was assumed that no initial vertical force was applied to the damper. Only the reaction forces of the damper in the vertical direction could occur by boundary conditions, due to the lateral loading.

These dampers are structurally simple, easy to design, and capable of dissipating all the seismic energy in multi-axis directions. However, the analytical models of this dampers were allowed to deflect only in 2-dimensional directions in this study. Thus, the out-of-plane deformations and buckling of the metallic damper were not considered. This limitation simplified the optimization process as much as possible, considering the complexity in the dynamic responses of the damper. Based on these postulations, this study focused on optimizing the initial rectangular shape of the proposed prismatic metallic damper.

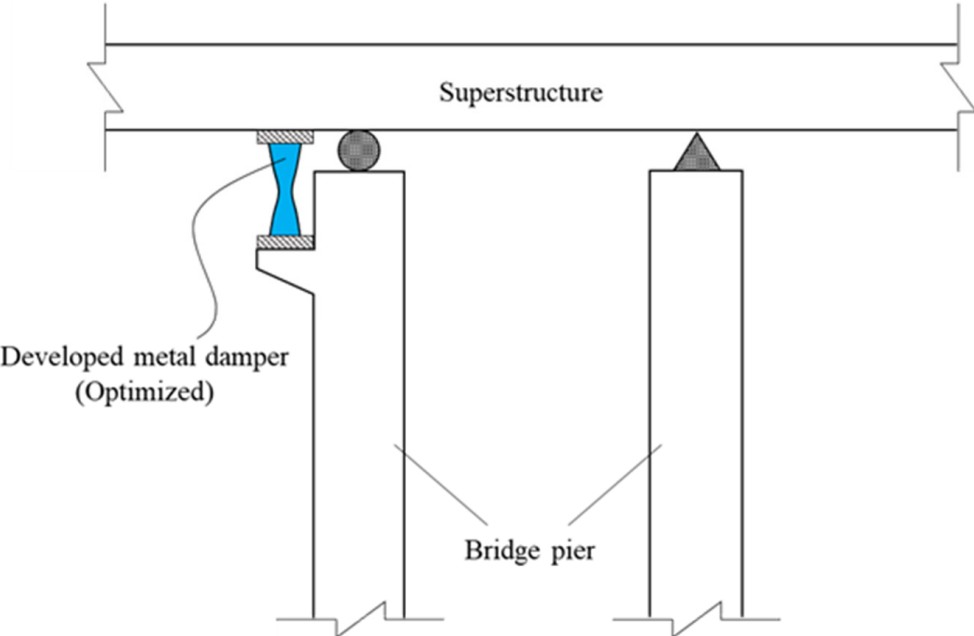

**Figure 1.** Conceptual installation of the proposed damper.

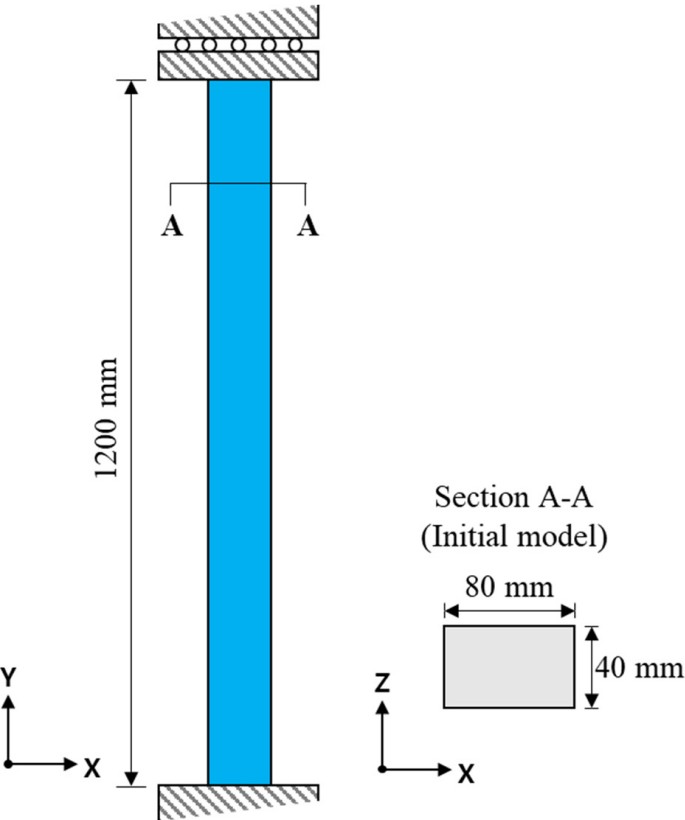

**Figure 2.** Initial geometry of the proposed damper.

### 2.2. Optimization Approach

The purpose of the optimization process was to find the optimum shape that dissipates the greatest amount of seismic energy during an earthquake. The optimization objectives are represented by Equation (1). The amount of dissipated energy is represented by $E(x_i)$. This energy term comprises the variables that are the control points, $x_i$, where $x_i$ indicates the width of the damper at location i. However, to obtain a single valid result through the process of optimization, boundary constraints were also applied. Here, two different boundary conditions were applied. First, the coordinates of the control points were regulated by their constraints. Second, the constraint of the total volume of the damper was determined. The first boundary condition restricts the alteration with the upper and lower bounds for the control point coordinates, $x_i^U$ and $x_i^L$, respectively. Moreover, the volume constraint was applied so that the volume of the changed shape might not exceed the initial volume. This constraint also helped to increase the efficiency of the device by using only a small amount of the component material.

$$
\begin{aligned}
\text{Maximize} \quad & E(x_i) \ (x_i : \textit{shape variables}) \\
\text{Subject to} \quad & V(x_i) \le V_0 \\
& x_i^L \le x_i \le x_i^U
\end{aligned}
\tag{1}
$$

The amount of energy dissipation is derived by extracting the maximum ALLPD from the ABAQUS FE analysis. ALLPD represents the energy dissipation through the plastic deformation of the model during the analysis. This value is calculated by using Equation (2):

$$
\text{ALLPD} = \int_V \int_0^t \sigma : \dot{\varepsilon}^{pl} \, dt \, dv
\tag{2}
$$

where $\sigma$ is the total stress and $\dot{\varepsilon}^{pl}$ is the plastic strain rate that occurred on the model volume $V$ during the analyzed time, $t$. The $\sigma$ and $\dot{\varepsilon}^{pl}$ are affected by the deformation of

the damper under the given loading conditions. Therefore, they are eventually affected by the initial un-deformed geometry of the damper, which is defined by the control variables $x_i$. Through alteration of the control variables, the optimization process was conducted by repetitive FE analysis. An integrated system comprising MATLAB R2018a and FE software (ABAQUS) was established. In this system, the geometrical alteration is carried out by modifying the control point coordinates in a Python script that runs on ABAQUS. After the analysis is finished for one changed shape, the Python script extracts the dissipated energy, which becomes the objective of the total optimization process. A complete flowchart of this process is shown in Figure 3.

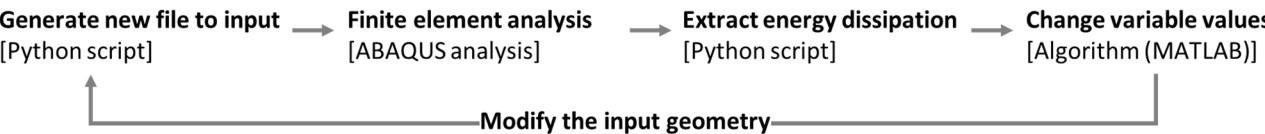

**Figure 3.** Flowchart of the optimization.

The sequential quadratic programming (SQP) algorithm was used for the optimization in MATLAB processing. The SQP algorithm is one of the constrained nonlinear optimization algorithms used by the MATLAB toolbox. The algorithm consists of three main steps: updating (1) the Hessian matrix, (2) the quadratic programming solution, and (3) the line search and merit function. The SQP algorithm executes all of the iterative steps in the region limited by the constraints, and if the constraint is not satisfied, the merit function is applied by combining the objective function and the constraint function into one gain function [30]. Its proven robustness makes the algorithm suitable for this problem, where it is difficult to derive an explicit relationship between shapes and object function due to the strong geometric nonlinearity.

*2.3. Static and Dynamic Optimization*

Since metallic dampers can be easily fabricated [24], they can be inexpensively produced with various geometries. Thus, this study attempted to make each optimized shape specific to different optimization conditions. These conditions represent the diverse installation environments of the subject metallic dampers. In other words, the dampers can be optimized with respect to the specific characteristics of individual structures.

Generally, a slope of the elastic region in the hysteresis loop becomes stiffer during the shape optimization process. It eventually contributes to an increased area of the hysteresis loop, which is equivalent to the energy dissipation. However, in the case of low loading conditions, such as mild earthquakes, an elastically stiffer damper might be unfavorable, as there is little deformation. According to the structural analysis conducted in this study, a novel topology of the optimized damper was found to be more effective with moderate ground accelerations with smaller peak ground acceleration and higher frequencies. To customize the damper shapes diversely for various expected installation conditions, several loadings with different maximum strokes or numbers of cycles were applied to the optimization process. Moreover, to overcome the limitation of general static optimization, the dynamic characteristics of the damper were considered. This advanced optimization process was conducted based on the dynamic FE analysis.

**3. Dynamic Analysis of Static Optimization**

*3.1. Design Variables*

During the optimization process, the geometrical shape of the damper was modified by the control points on the free boundary. In order to set the design parameters, several pre-analyses were carried out to determine the control points. In a previous study [31], all the control points were set to be located at the point on an outer edge of the member. In addition, the change in these designated variables was applied inversely to the coordinates of the geometrically symmetric points. A new shape was generated by splines connecting

all the points that extruded with a fixed thickness according to the sketch functions of ABAQUS. Thus, the generated shape was symmetrical from the perspective of up–down and left–right. The results of this previous study could be summarized as follows. In order to reduce the calculation time and to simplify the optimal curve, the three widthwise variables in the plane satisfy both efficiency and validity. These three variables are at the end, the quarter, and the mid-points of the member, respectively, in the longitudinal direction. The effects of the height variables, which move longitudinally to the member, were not significant.

Therefore, this present study follows the 3-variable configuration of the preceding work. As illustrated in Figure 4, the variables $x_b$ (boundary), $x_c$ (chest), and $x_w$ (waist) determine the $x$-directional width.

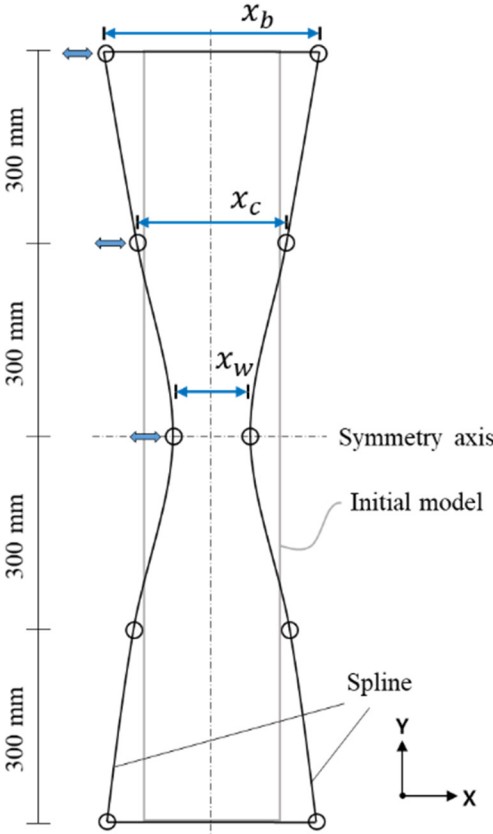

**Figure 4.** Specifications of the control points.

### 3.2. Finite Element Model

FE analysis was conducted for each iteration to calculate the changing amount of energy dissipation resulting from the change in shape. The design conditions were assumed to be as simple as possible for this optimization. Generalized properties were selected with reference to ASTM A36 steel, which is a common mild steel. The yield strength was 250 MPa, the Poisson ratio was 0.26, and the modulus of elasticity was 200 GPa. The material was also assumed to have completely plastic behavior. This elementary constitutive model was determined to simplify the optimization problem, considering the complex loading conditions of the proposed optimization problem, such as dynamic excitation, which will be discussed later. Eight-node brick elements (C3D8) were used, and the failure of the material was not considered. The mesh size was approximately 10 mm, and the total number of elements was roughly 5000. These values were determined through initial parameter studies and set to demand as little computational cost as possible in providing data independent of the mesh size. The shape optimization was based only on the correspondence to a uniaxial seismic load.

Under vertically fixed conditions, a cyclic load was applied horizontally to the top of the prismatic metallic damper. Through this assumption of the loading condition, the developed damper worked in a way similar to general shearing dampers, such as shear slit dampers and shear panel dampers. The cyclic loading scheme was controlled with various conditions for several cases with different strokes. In total, there were four cases with different scales of multi-cycled strokes (Figure 5). Following a previous study (Deng et al., 2014), each case had nine cycles of strokes, which was determined by considering the computational speed. Conventionally, a loading scheme is utilized by gradually increasing the displacement in a single multi-cycle condition. However, through these various loading schemes, it was possible to determine whether the optimum shape varied with respect to the required capacity.

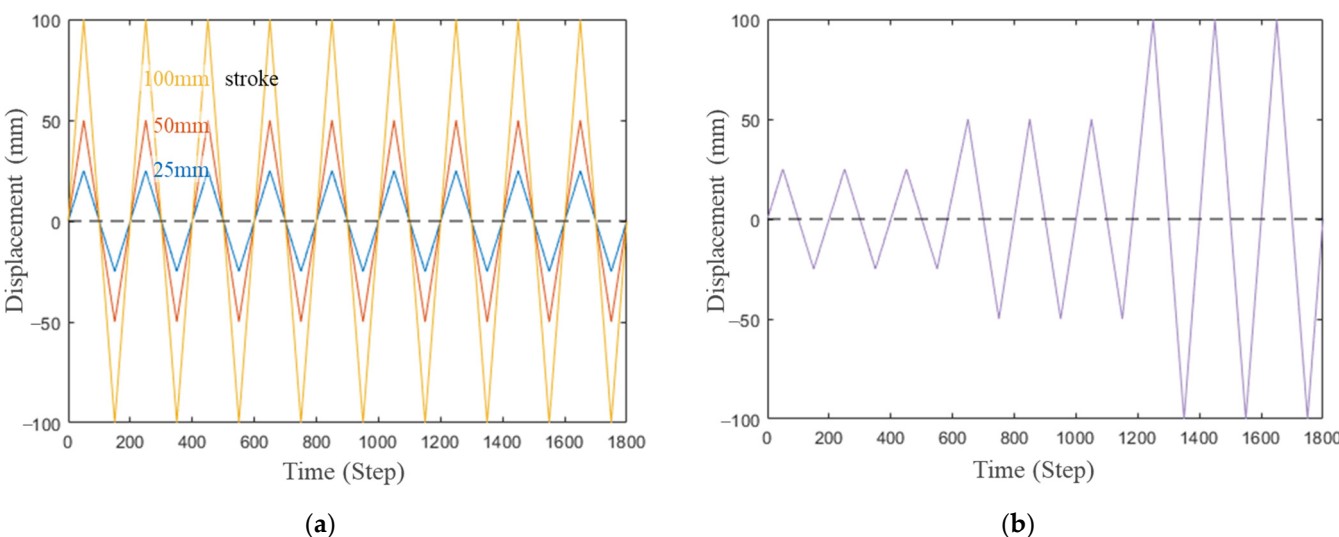

**Figure 5.** Simplified static loading scheme for optimization: (**a**) Three cases with a uniform scale of strokes; (**b**) Additional case with a stroke increase.

Here, the loading conditions were controlled with a static load, i.e., the displacement. However, to evaluate how the optimized damper will behave under real dynamic circumstances, the model should be considered with the acceleration. Therefore, further optimization cases with dynamic FE analyses were conducted. The dynamic optimization process also followed the general optimization process described in this section, with some changes to the boundary and the loading conditions. The specific modifications are explained in Section 4.1.

### 3.3. Static Optimization Results

The optimized shapes were derived under four different stroke conditions. The optimization history curves of each case are illustrated in Figure 6a. The overall optimized results of the three variables are provided in Figure 6b and Table 1. The cases are named S1–S4, corresponding to the four stroke conditions. The optimization stroke conditions of each case are specified in the table, and the e volume and increased stiffness of the damper in the elastic region of the optimized dampers are also presented. The results of the dissipated energy and the improvement of each case were declared with the stroke conditions of S4, i.e., nine cycles with an increase in strokes. This unification of the loading conditions was applied to determine the relative evaluation of the results.

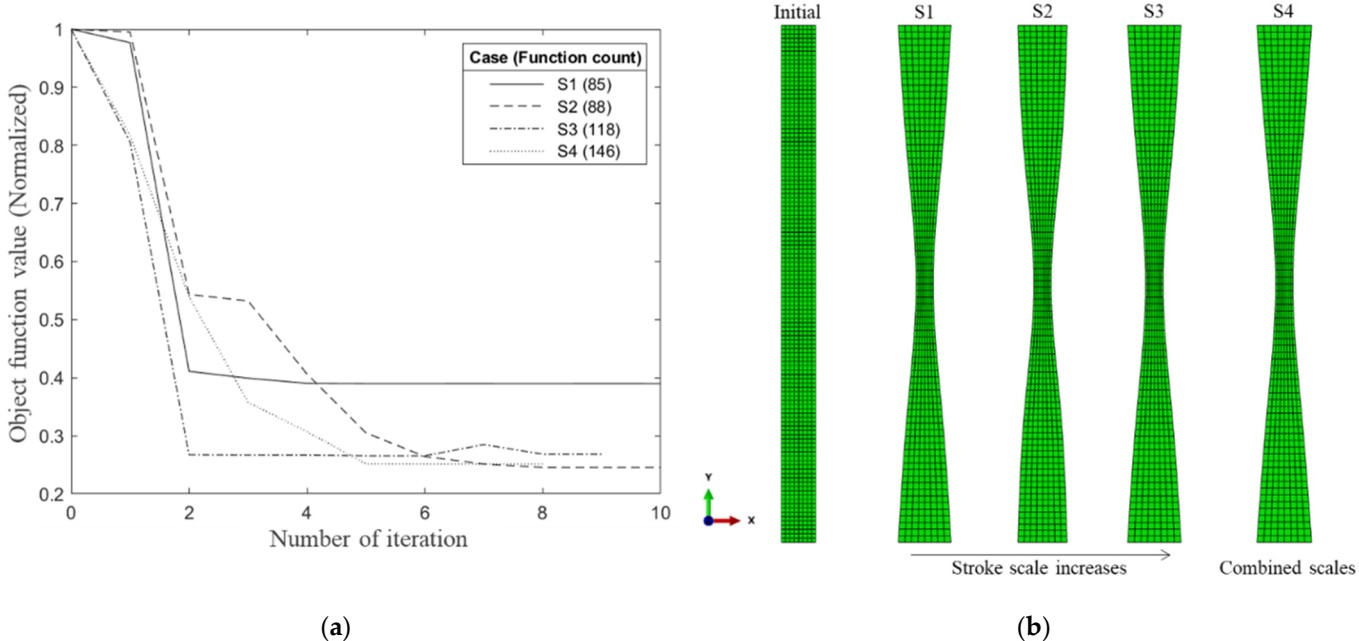

(**a**)            (**b**)

**Figure 6.** Static optimization results: (**a**) Optimization history curves of cases S1–S4; (**b**) Optimized shapes with static analysis (S1–S4).

**Table 1.** Shape variable values, energy dissipation, and improvement of each optimized model with a fixed thickness.

| Cases | Initial | S1 | S2 | S3 | S4 |
|---|---|---|---|---|---|
| Optimized Stroke size | | 25 mm | 50 mm | 100 mm | Max. 100 mm * |
| $x_b$ [mm] | 80 | 123.37 | 113.87 | 123.23 | 126.61 |
| $x_c$ [mm] | 80 | 81.34 | 84.60 | 81.42 | 80.10 |
| $x_w$ [mm] | 80 | 40.00 | 40.00 | 40.00 | 40.00 |
| Volume $[\text{m}^3]$ | 3.840 | 3.840 | 3.839 | 3.840 | 3.836 |
| Elastic stiffness [MN/m] | 2.314 | 3.705 | 3.730 | 3.709 | 3.672 |
| ALLPD [kN·m] | 53.76 | 93.99 | 91.01 | 93.91 | 93.64 |
| Improvement [%] | | +74.9 | +69.3 | +74.7 | +74.2 |

* Case S4: The case with stroke increase.

Figure 7 compares the stress distribution of the final optimized model (S1) at the edge, with the initial shape of the damper. This example results from the final 50 mm stroke-optimized model. The plotted graph shows the von Mises stress at each element located at the edge of the model. The dotted line indicates the stress distribution at the edge of the model with the initial geometry, and the solid line denotes the optimized geometry. The solid line of the optimized model proves that the stresses are well distributed, and more elements are in a plastic state than they were in the initial model. The number of elements at the edge that reached the plastic state showed an increase of approximately +180%. This result of the optimization process eventually increased the hysteresis loop area (Figure 8).

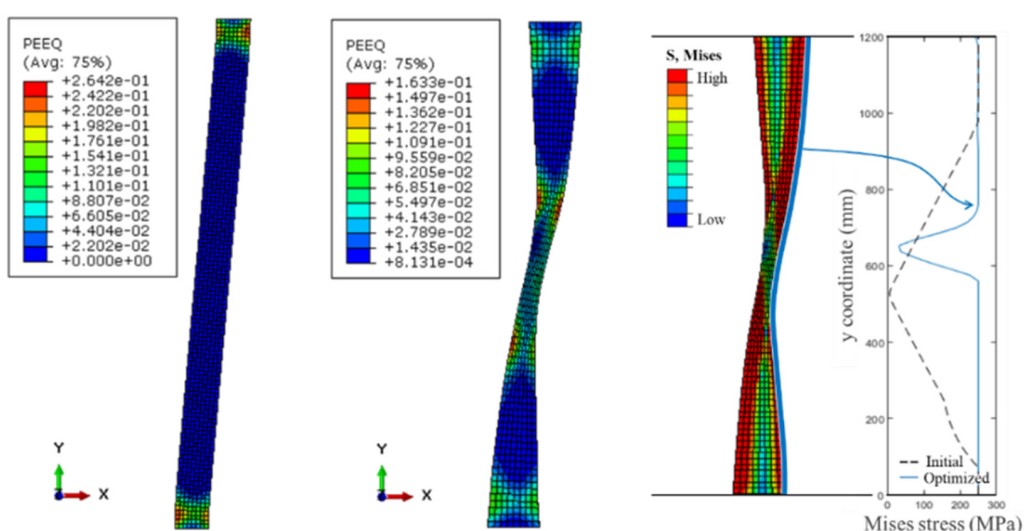

**Figure 7.** Plastic strain distribution (initial and S1) and stress distribution at the edge (S1).

(**a**)

(**b**)

(**c**)

(**d**)

**Figure 8.** Hysteresis loops of the optimized shapes: (**a**) Case S1; (**b**) Case S2; (**c**) Case S3; and (**d**) Case S4.

All the optimized damper shapes showed significant improvement in energy dissipation, with every case demonstrating 70–75% improvement. Through the shape optimization, the width of the top and bottom ends became larger and that of the middle became smaller. This result of the optimization process increased the number of elements in a plastic state and eventually enhanced the hysteresis loop area (Figure 8). The hourglass shape was predictable and was already derived by a number of other researchers [1,13,29].

However, variety in the optimization results with respect to changes in the loading conditions was scarcely observed. The results were very slightly diversified only in the case of S2, which was under a repetitive 9-cycle 50 mm stroke. Other loading conditions, which were 9-cycle 25 mm (S1), 9-cycle 100 mm (S3), and 9-cycle with a stroke increase (S4), led to similar shapes. There were only vague correlations between the geometrical changes of the optimized results and the increases in stroke size. The minor differences between the cases, including case S2, can be considered as the results of the local minimum.

Therefore, it can be concluded that it is difficult to find specialized shapes for specific loading conditions using optimization with static analysis. Furthermore, the optimized shapes are presumed to have certain limitations under particular dynamic conditions. On the other hand, these limitations of the static optimization process can be overcome through a dynamic optimization process, which is described in the next section.

### 3.4. Dynamic Analysis of the Statically Optimized Damper

The optimized shape of the presented damper is derived from the optimization process with a static analysis using an approach that is similar to previous studies [13,14,22,29]. However, this approach can have some limitations. As stated in Section 3.3, it is difficult to induce specialized shapes for the different capacities required. The control of the stroke size or cycles scarcely influences the optimized shape. Moreover, since the shape-optimized damper becomes stronger and stiffer than the initial model, the plastic deformation of the damper can decrease with a small amount of ground acceleration. Because of this reduction of deformation, the optimized damper might dissipate less seismic energy during small earthquakes.

To consider the dynamic characteristics of a metallic damper, a dynamic analysis was conducted through ABAQUS to examine the capability of the previously optimized damper under dynamic loading. In order to evaluate the tendency as simply as possible, a prototype bridge model was constructed (Figure 9). This structural model was composed of two prototype piers and one prototype superstructure in a "simply supported" condition. The structure was assumed to be excited through the ground, only in x-direction (in-plane). The lengths of the piers and the superstructure were 10 m and 30 m, respectively. The same material, i.e., steel, was used, which was applied in the modeling of the damper, with both the prototype column and superstructure having the same circular cross section with a diameter of 0.5 m. Thus, the mass of the super structure was 49.45 ton, and the first natural frequency of the structure was 1.23 Hz. Here, the total lateral stiffness of the bridge was mainly due to the stiffness of the prototype columns. The intrinsic damping ratio of the material was presumed to be zero, in order to ensure that the dissipation of seismic energy occurred only through the plastic deformation of the damper. Because this prototype bridge model did not consider the structural design aspect, the self-weight of the structure was disregarded.

To analyze the tendency of the optimized damper's behavior with respect to the scales of the ground motion, two simple ground accelerations were applied to the model. The accelerations were presumed on the basis of the sine functions of one cycle with two maximum accelerations: 0.1 g and 0.2 g (where g indicates the gravitational acceleration) having a frequency of 0.5 Hz. The sine function with a maximum of 0.1 g represents a small earthquake, and the sine function with a maximum of 0.2 g represents a larger earthquake. The FE analysis was conducted using dynamic-implicit time integration in ABAQUS with default settings. This time integration scheme uses a Hilber-Hughes-Taylor operator, which is an extension of the Newmark β-method [32,33]. In ABAQUS, the operator utilizes a very

slight numerical damping as a default value to enhance the numerical stability, but the damping effect of the integration scheme was almost negligible in this study.

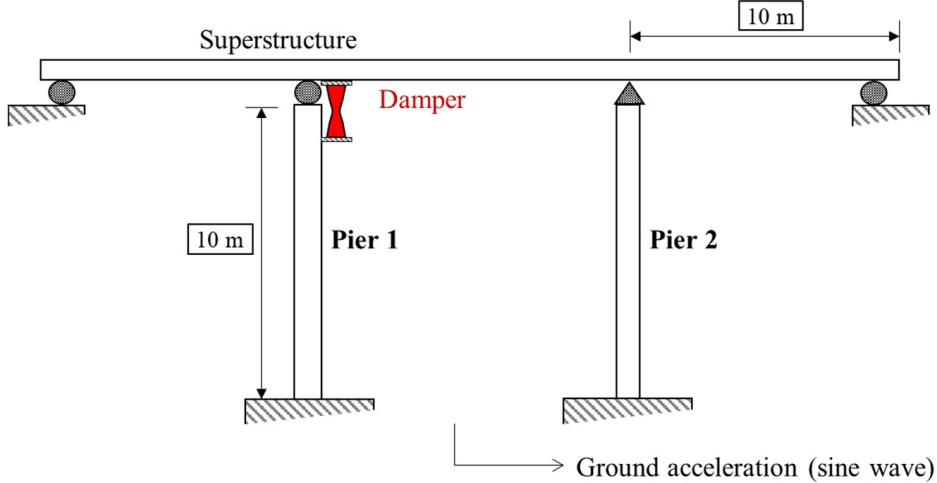

**Figure 9.** Prototype bridge dimensions and boundary conditions.

The results of the analysis are provided in Table 2. The initial shape and one optimized case are compared. Case S1 represents the results of the static optimization without the thickness variable, due to the similarities between the complete results. As in the previous static, the optimized damper dissipates more energy than the initial shape damper in the case of the larger 0.2 g acceleration. However, the plastic energy dissipations of S1 decrease at a smaller acceleration of 0.1 g. This result confirms that the optimized damper will dissipate less energy for small ground acceleration with high probability. Therefore, further optimization-aimed dynamic modeling was constructed to overcome this limitation of the static optimization approach.

**Table 2.** Energy dissipation of each optimized shape with a structure under dynamic loading.

| Structural Model Analysis | Initial | Optimized (Case S1) | Initial | Optimized (Case S1) |
|---|---|---|---|---|
| | Dynamic Analysis (Max. 0.1 g) | | Dynamic Analysis (Max. 0.2 g) | |
| ALLPD [kN·m] | 4.82 | 4.61 | 18.91 | 21.27 |
| Rate of change | | 4.42% | | +12.5% |

## 4. Dynamic Optimization

### 4.1. An Analytical Model for Dynamic Optimization

The prototype bridge model presented in the section above was created to illustrate the possible tendency of the damper's behavior under dynamic loading. Many more aspects of the model need to be reconsidered, such as its precise dimensions and properties, in order to represent a real structural environment for a damper. However, it is impossible to utilize a realistic structural model in the optimization process, since the computational cost will be excessively demanding. Thus, an analytical model is proposed in this section for the dynamic optimization process. This model is based on a concept similar to that of the preceding model. The overall configuration of this analytical model is explained in Figure 10.

Both ends of the damper are only allowed to move in one lateral direction with fixed movements. This dynamic model contains two additional components—the concentrated mass and the spring set. These components are also applied to each end of the member, representing the complete structure on which the damper will be installed. Several sets of values for the stiffness and concentrated mass that could be matched to a specific structure

can be considered. The stiffness of spring set 1 and spring set 2 is calculated from the expected stiffness of the prototype column, pier 1 and pier 2 in Figure 9. The point mass of the upper end is derived by considering the mass of the superstructure. The other mass at the lower end originated from the mass of the upper portion of the pier where the damper was installed. Therefore, for the bridge structures, the concentrated mass at the upper end is larger than the other mass at the lower end. Here, a spring stiffness of 1840.78 kN/m, and 50 and 4 tons of concentrated mass, were applied to match the previously presented prototype structure. The analysis result of the simplified model with the parameter assumptions was compared with that of the target structure, i.e., the prototype bridge. The comparison was conducted utilizing the initial geometry of the damper. The responses of both cases, which were relative lateral displacements between the upper and lower ends of the damper, were well matched (Figure 11).

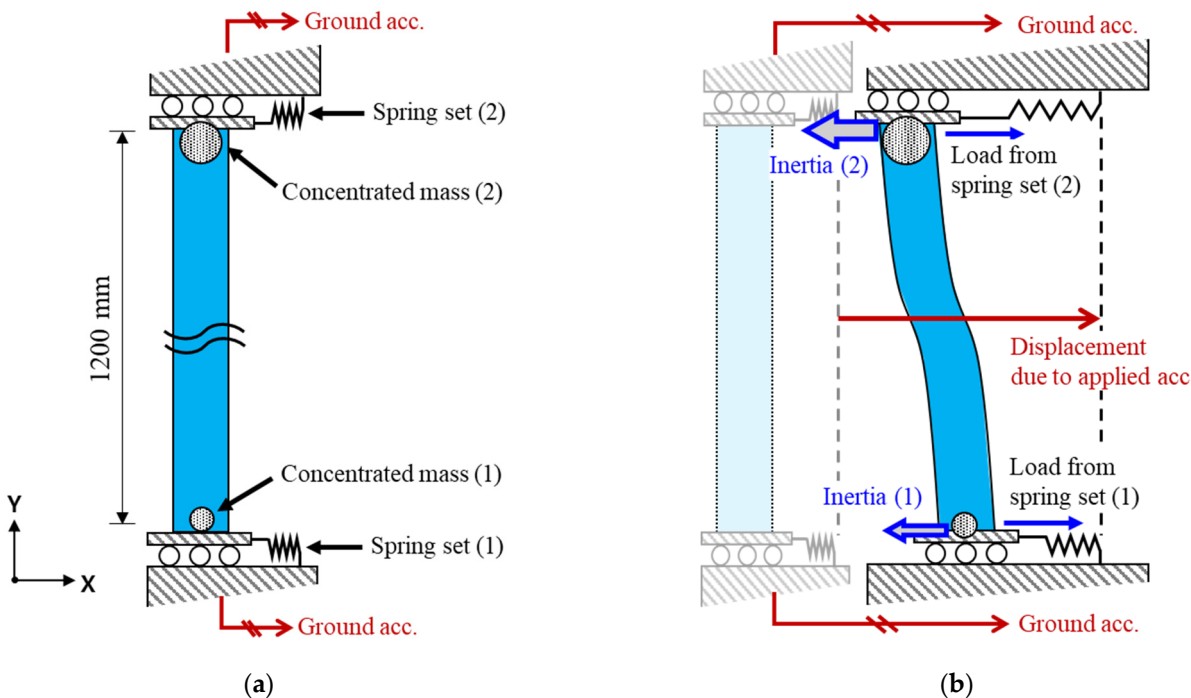

**Figure 10.** Simplified model for dynamic optimization: (**a**) Model configuration; (**b**) Deflection of the model under the applied excitation.

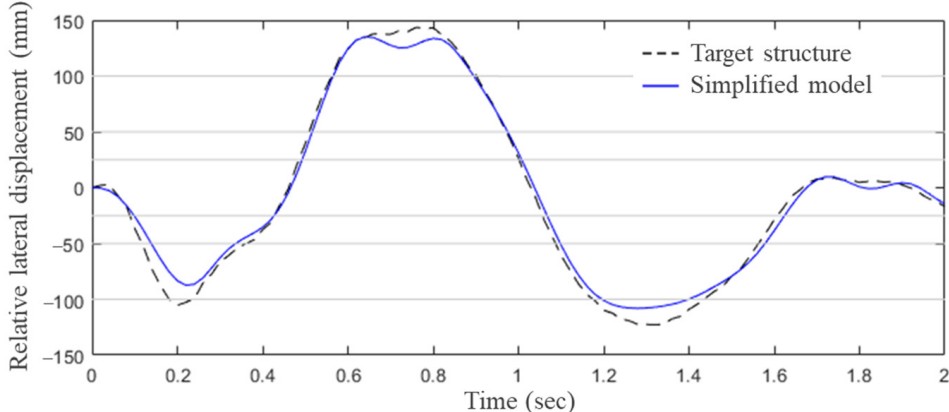

**Figure 11.** Specifications of the control points. Relative lateral displacement between the upper and lower ends of the damper under sine-function loading with maximum 0.2 g acceleration.

An evaluation of the optimized dampers was conducted again, using the dynamic analytical model. As before, the energy dissipation capacities of the initial shape and the

two optimized shapes were compared, with two different maximum ground accelerations. A tendency to closely match the previous structural analysis was confirmed. The optimized shapes showed less energy dissipation under small ground accelerations. The scale of dissipation decreased to a smaller degree compared to the total structural analysis. This was because the overall displacement of the analytical model was slightly smaller due to a minor intrinsic error in the model. However, the utilization of this model is still valid for design purposes, since it has conservative assumptions.

Moreover, a detailed analysis was carried out regarding specific statically optimized cases. Case S1 was examined as the representative case. It was confirmed that the optimized shape underperformed compared to the initial shape, until a certain amount of ground acceleration was reached. In this specific case, the energy dissipation capability of the optimized shape S1 lagged behind the initial shape, up to around 0.14 g (Figure 12).

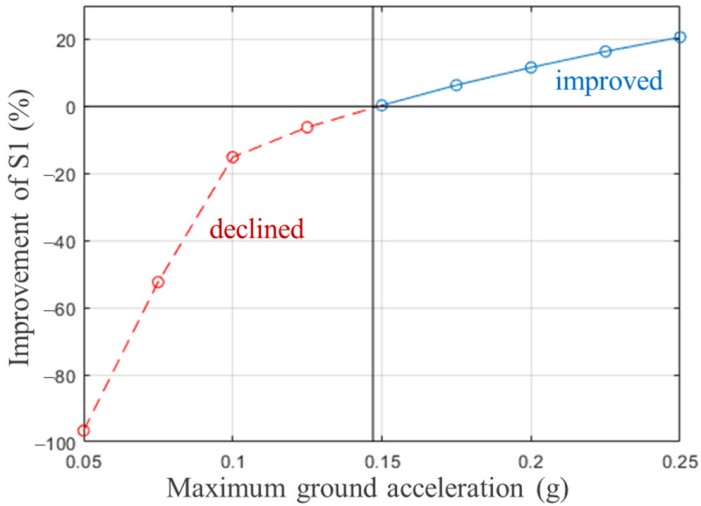

**Figure 12.** Improvement of case S1 in energy dissipation from the initial shape analyzed with the dynamic optimization model.

*4.2. Dynamic Optimization Results*

Dynamic optimization was accomplished with the three variables ($x_b$, $x_c$, and $x_w$). A total of nine maximum ground acceleration cases were tested, from 0.05 g to 0.25 g. A frequency of excitation, 0.5 Hz, was first applied, which was the same as that of the previous structural dynamic analysis in Section 3.4. The optimization results and improvements are shown in Fiugres 13 and 14a and Table 3. The plastic energy dissipation increased in all the cases compared to those of the initial and statically optimized case (S1). The improvements from case S1 were significant for small acceleration and decreased with an increase in the given acceleration scale (Figure 14a). It should be noted that extremely high improvement was obtained for smaller acceleration. This is because the statically optimized shape scarcely reaches plastic deformation at this level of acceleration. For comparison of a dynamically optimized case (D3) with the static optimization result (S1), the expected hysteresis loops are shown in Figure 14b under a sine function loading. The percentage displayed on the graph is the rate of change of the ALLPD values compared to the basic model before optimization. In this case, it can be considered as the area of the hysteresis loop. Under the small acceleration (Maximum 0.1 g), the energy dissipation of case S1 decreases, whereas D3 dissipates more energy.

There were significantly different trends between the results under small and large acceleration conditions. The optimized values of the variable $x_b$ were larger than $x_c$ at high acceleration, a result that is similar to the static optimization results. On the other hand, the optimized value of $x_b$ was lower than $x_c$ at low acceleration. The transition of this trend coincides with the inflection point in Figure 12 from negative to positive improvement of

the dissipation capability of the statically optimized shape. The value of the variable $x_w$ was independent of the change in ground acceleration scales.

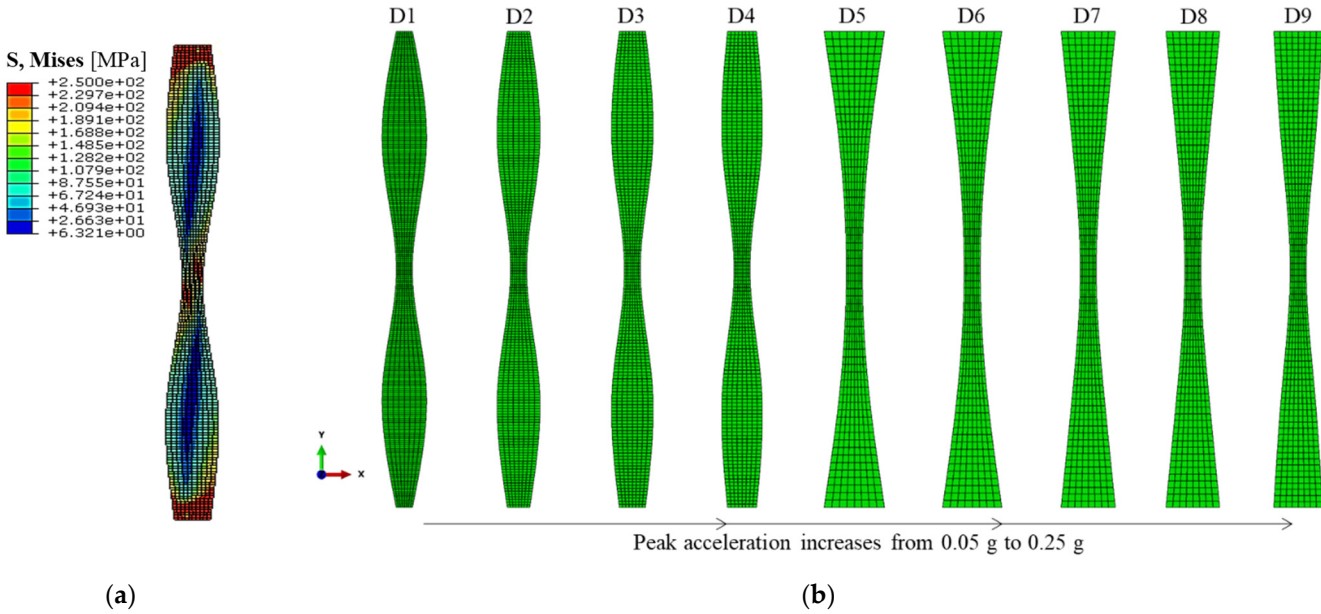

(**a**)

(**b**)

**Figure 13.** Dynamically optimized shapes under various acceleration conditions: (**a**) Representative stress distribution of the novel topology (Case D3); (**b**) Optimized results of all cases.

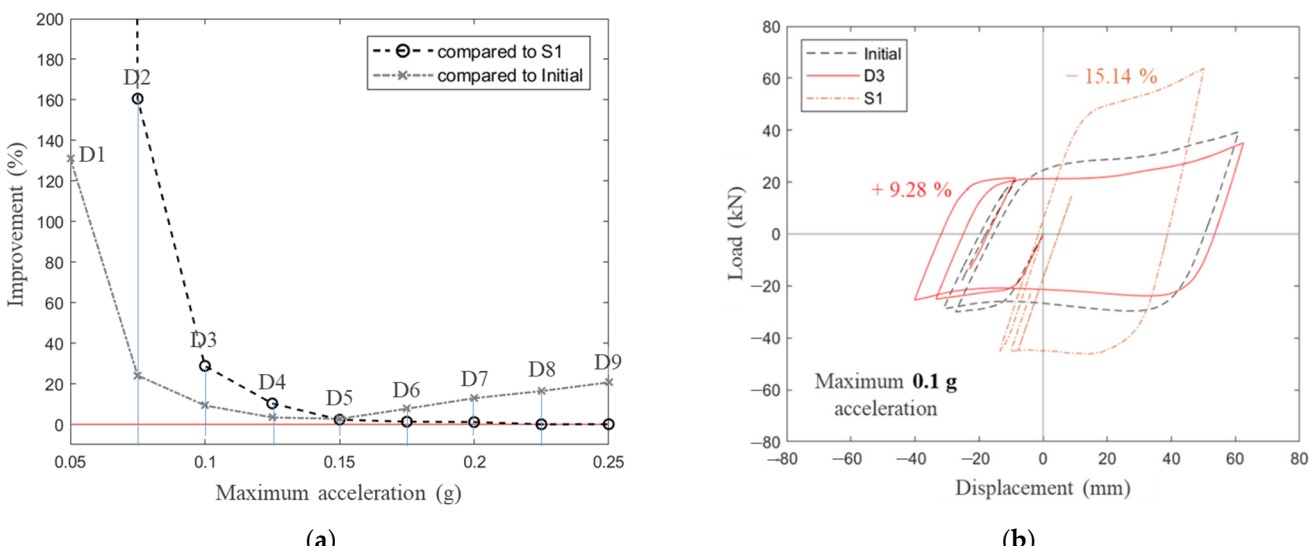

(**a**)

(**b**)

**Figure 14.** The energy dissipation of the dynamically optimized dampers: (**a**) Improvement of results; (**b**) Hysteresis loops of S1 and D3.

In addition, the effect of the number of excitation cycles on the dynamic optimization was analyzed. As mentioned earlier, in order to reduce the computational cost of the iterative operation conducted by the optimization algorithm, only one cycle of sine wave was first applied as the loading condition. However, considering that a more stationary response is required for the actual loading such as an earthquake, a new set of the dynamic optimization results was derived. In this case, the total time period of the excitation was extended to 8 s to increase the number of sine-wave cycles four times. The derived optimization variable values are shown in Figure 15b. Consequently, the changing trend of the optimized variable values due to the differences in the peak acceleration was more stably revealed, but the point at which the trend changes was maintained, the same as in

the case of a single cycle. Therefore, it was found that the one-cycle condition was sufficient to see the macroscopic changes in the trend of the dynamic optimization results. Based on this result, the effect of excitation frequencies was analyzed as in the next section.

**Table 3.** Shape variable values, energy dissipation, and improvement of each optimized model with a fixed thickness.

| Case | D1 | D2 | D3 | D4 | D5 | D6 | D7 | D8 | D9 |
|---|---|---|---|---|---|---|---|---|---|
| Max. acc. | 0.05 g | 0.075 g | 0.1 g | 0.125 g | 0.15 g | 0.175 g | 0.2 g | 0.225 g | 0.25 g |
| $x_b$ [mm] | 40.46 | 55.86 | 65.32 | 73.05 | 152.15 | 149.51 | 137.04 | 135.42 | 123.19 |
| $x_c$ [mm] | 109.84 | 104.56 | 101.31 | 98.66 | 71.44 | 72.17 | 76.62 | 77.20 | 81.36 |
| $x_w$ [mm] | 40.00 | 40.00 | 40.00 | 40.00 | 40.03 | 40.47 | 40.00 | 40.00 | 40.08 |
| Volume $[\text{m}^3]$ | 3.839 | 3.840 | 3.840 | 3.840 | 3.839 | 3.835 | 3.839 | 3.840 | 3.838 |
| Elastic stiffness [MN/m] | 1.277 | 2.105 | 2.567 | 2.890 | 3.264 | 3.305 | 3.554 | 3.580 | 3.703 |
| ALLPD [kN·m] | 0.96 | 2.35 | 4.38 | 6.97 | 10.08 | 13.98 | 18.42 | 23.39 | 29.01 |
| **Improvement** | | | | | | | | | |
| to Initial | +131.1% | +24.1% | +9.3% | +3.4% | +2.7% | +7.7% | +12.9% | +16.4% | +20.7% |
| to S1 | +6762.0% | +160.6% | +28.8% | +10.3% | +2.3% | +1.3% | +1.1% | +0.01% | +0.07% |

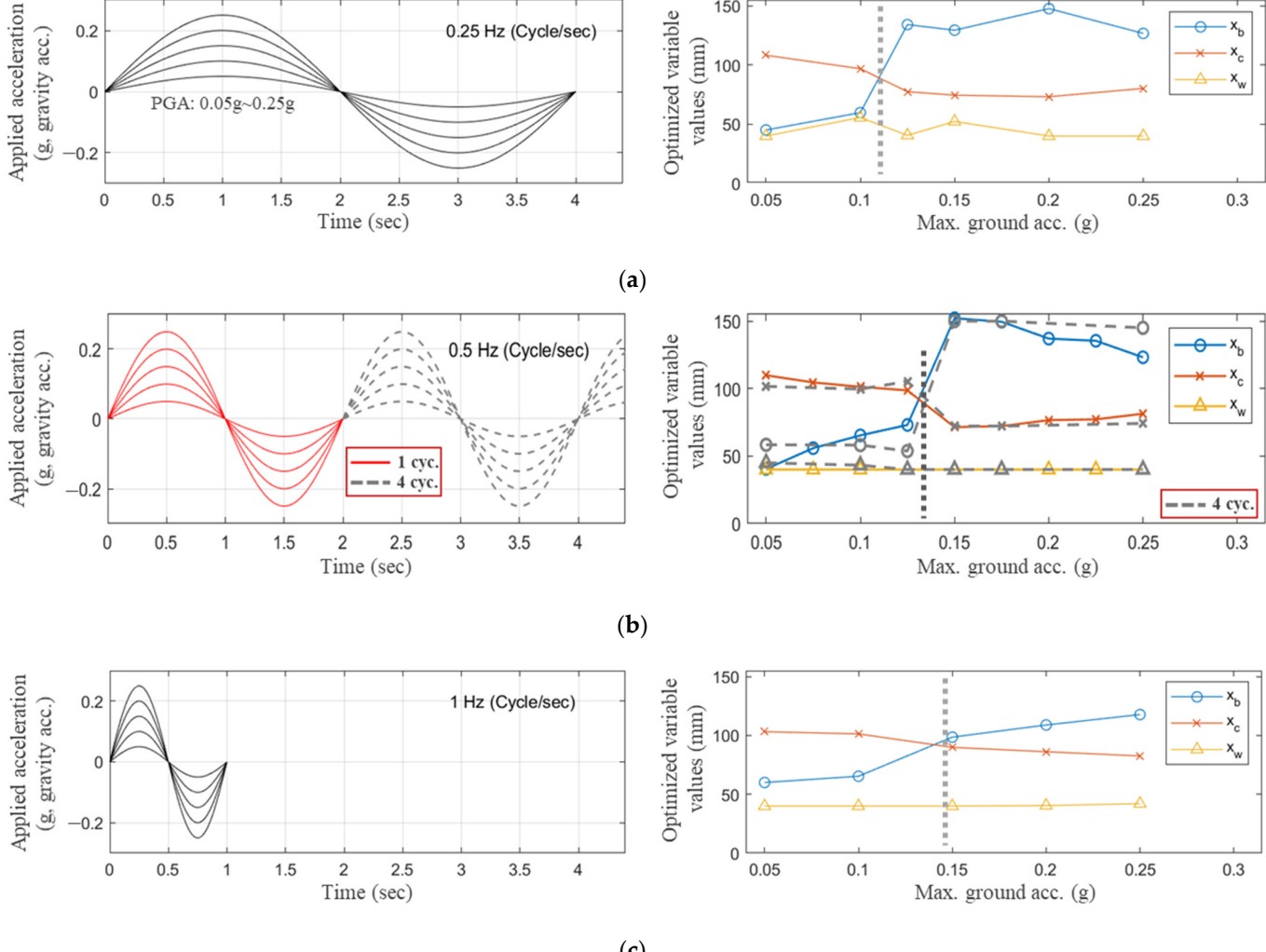

(**a**)

(**b**)

(**c**)

**Figure 15.** *Cont.*

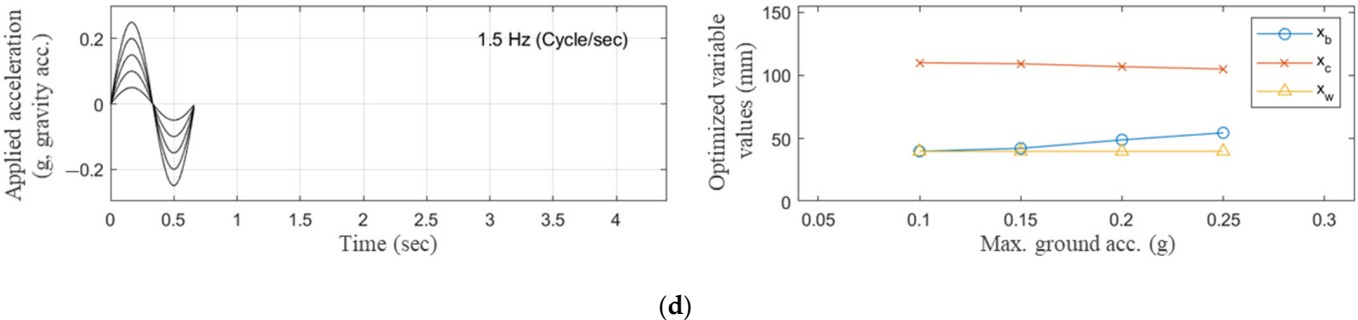

**(d)**

**Figure 15.** Dynamic optimization results with respect to the frequency of applied excitation: (**a**) 0.25 Hz; (**b**) 0.5 Hz; (**c**) 1 Hz; and (**d**) 1.5 Hz.

*4.3. Effect of the Excitation Frequencies to the Dynamic Optimization*

The change in the trend of these optimization variable values was affected not only by the scale of peak acceleration, but also by the frequency of the applied sine excitation. Figure 15 illustrates the optimized variables when the dynamic optimization is performed by increasing the frequency of applied excitation from 0.25 Hz to 1.5 Hz. Here, the values in Figure 15b are the same as those in Table 3 with the frequency of 0.5 Hz. Through the results with respect to the acceleration frequency, the peak acceleration value can be compared at the point where the trend of optimized geometry changes as the frequency increases.

In the previous optimization results at 1 Hz, the trend changed between 0.125 g and 0.15 g, which are the peak values of applied acceleration (Figure 15b). However, at a lower frequency (0.25 Hz), the trend changed at a smaller peak acceleration (Figure 15a). In contrast, when the frequency was higher (1.0 Hz), the peak acceleration value that changes the trend became larger (Figure 15c). Further, when the applied frequency exceeded 1.5 Hz, only one trend of the optimized shapes was derived in all cases between the given peak acceleration value (0.1~0.25 g), which was the newly derived optimal shape suitable for the small earthquake (Figure 15d).

Therefore, it can be concluded that the novel shape derived through dynamic optimization will dissipate energy more effectively in a relatively moderate earthquake. Here, a moderate earthquake can be defined as an earthquake with low seismic energy, which means an earthquake with a smaller peak ground acceleration and higher frequencies. In order to examine the behavioral characteristics of the derived damper shapes more thoroughly, virtual excitations with various frequencies and accelerations similar to an actual earthquake were introduced in the next section, by further exploring the simple sine waves applied as the conditions of the dynamic optimization.

*4.4. Seismic Analysis of the Optimized Results*

Cases D3 and D7 are representative of the two optimized tendencies under small and large ground acceleration, respectively. Case D7 was wider at both ends, which is analogous to the static optimization results. Likewise, the static hysteresis loop area also increased. In contrast, under low acceleration, the initial shape converged into a different form. The widths of both of the ends were smaller than that of the statically optimized shape. Consequently, the damper became slightly weaker than the initial shape, which resulted in an increase in the dissipation of the plastic energy.

To evaluate and verify the performance of the optimized dampers, seismic analysis of the two representative dampers was conducted (Figures 16 and 17). The damper shapes of D3 and D7 were examined, with the same conditions as the dynamic optimization model. The boundary conditions, including the spring set and the concentrated mass, were maintained. However, in this case, artificial earthquake acceleration was applied instead of sine-function acceleration (Figure 16a). Artificial acceleration was created randomly with the design response spectrum. The construction of the response spectrum followed the

Uniform Building Code [34] (Figure 16b). The ground coefficients $C_a$ and $C_v$ were identical at 0.09, which were the design spectrum accelerations for a single period and 1 s.

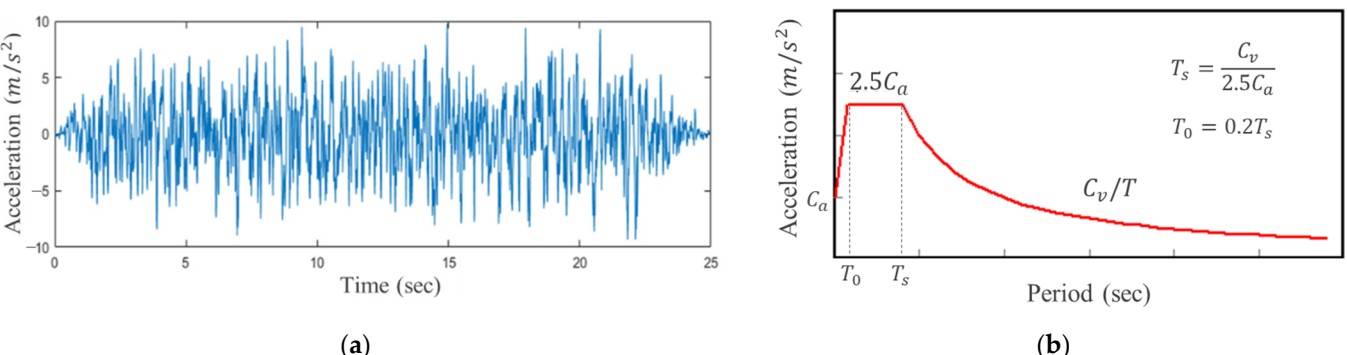

(**a**)                                                            (**b**)

**Figure 16.** Formation of an artificial earthquake acceleration: (**a**) Artificial earthquake accelerogram; (**b**) Design response spectrum.

(**a**)

(**b**)

(**c**)

**Figure 17.** Relative lateral displacement between the upper and lower ends of the damper under artificial earthquake acceleration (Cases D3 and D7): (**a**) Maximum 0.1 g; (**b**) Maximum 0.2 g; (**c**) Maximum 0.3 g.

The applied acceleration was scaled to match the intended peak ground accelerations, which were 0.1 g, 0.2 g, and 0.3 g. Comparing the plastic dissipation (ALLPD) between the two cases, case D3 surpassed the result of case D7 at the maximum 0.1 g acceleration condition. However, the amounts of dissipation became analogous to each other at the maximum 0.2 g condition, and the gap between the cases turned around at the highest acceleration condition. As illustrated in Figure 17, D3 deforms more than D7 at small accelerations. This can be interpreted as D3 having a larger applied stroke, which leads to larger energy dissipation. This apparently indicates that D3 will dissipate larger seismic energy under small-scale earthquakes. This trend of results under a given earthquake accelerogram equals that of the optimization condition, the sine-function acceleration. Thus, it can be rationalized that the previously proposed simple loading condition is sufficiently applicable for dynamic optimization.

## 5. Conclusions

This study proposed a novel methodology to develop a metallic damper coupled between a bridge pier and the upper part of a bridge. This study focused on enhancing the energy dissipation capability of the metallic damper with respect to different target capacities. Simplified finite element modeling and an optimization method were proposed, using finite element analysis and the SQP algorithm. First, optimization was performed with FE static analysis by applying various static load conditions to determine the optimal shape. Due to the limitation of the static optimization, further study based on dynamic analysis was completed. It was confirmed that optimized shapes for different earthquakes could be derived through this dynamic optimization process. The main conclusions obtained in this study can be summarized as follows.

- The optimized shape obtained through the static analysis showed an hourglass shape in which the width of both ends was increased, and the width of the center was reduced. This trend was similar to the results of previous studies. However, in the static analysis, it was difficult to determine the different optimal shapes for all stroke sizes, or to determine an appropriate virtual stroke for optimization.
- Through optimization with static analysis, the stiffness of the damper increased. Accordingly, the optimized shape dissipated a large amount of energy when the damper was deformed due to large earthquakes. In the case of small earthquakes, however, the optimized damper may undergo less deformation and dissipate only a small amount of energy. Identical results were obtained using the structural prototype bridge FE model.
- Thus, further optimization based on a dynamic analysis was carried out to determine the optimal shapes that dissipate the maximum amount of energy during relatively small earthquakes. To generalize the optimization method based on the dynamic analysis, an ideal analytical model was proposed with the spring set and concentrated mass.
- Through the proposed analytical model, an additional optimized shape for small earthquake accelerations was developed. The optimized shape at low acceleration and high frequency exhibited a slight reduction in width at both ends, unlike previous results. The new shape could lead to a greater amount of energy dissipation for small seismic loading.

Based on these findings, further studies will be conducted regarding practical applications. The analysis process of this study during the dynamic optimization procedure had been fairly simplified. Therefore, in the future research, a study of how to select the most efficient object function under various earthquake excitations—including energy dissipation, effective stiffness, or structural displacement—will be conducted. More realistic target structures will be applied as the optimization conditions, including an analysis of resonance with respect to their natural frequencies and intrinsic damping. Furthermore, experimental research will be conducted, along with the realistic conditions for the validation of the optimization.

**Author Contributions:** Conceptualization, J.W.P. and Y.M.L.; methodology, G.-H.Y. and J.-H.Y.; software, G.-H.Y.; validation, J.W.P. and Y.M.L.; formal analysis, J.W.P. and J.-H.Y.; investigation, J.W.P.; resources, J.-H.Y.; data curation, J.W.P.; writing—original draft, J.W.P.; writing—review and editing, G.-H.Y. and Y.M.L.; supervision, G.-H.Y. and Y.M.L.; project administration, Y.M.L.; funding acquisition, Y.M.L. All authors have read and agreed to the published version of the manuscript.

**Funding:** This research was funded by the National Research Foundation of Korea (NRF) grant number NRF-2019R1A2C1090426 and the APC was also funded by the NRF.

**Acknowledgments:** This research was supported by the National Research Foundation of Korea (NRF) grant funded by the Korea government (MSIT) (NRF-2019R1A2C1090426).

**Conflicts of Interest:** The authors declare no conflict of interest.

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
