# Peer review of "Effect of Dynamic Loading Conditions on Maximizing Energy Dissipation of Metallic Dampers"

_applsci, doi:10.3390/app12063086_

Round 1

Reviewer 1 Report

I have attached my review report for your information.

Author Response

Thank you for the important comments. Please see the attachment.

Reviewer 2 Report

The paper can be accepted for publication provided a revision is made. The following comments should be considered when revising the paper:

  • No mesh sensitivity study was carried out. In previous works, it was demonstrated that the results (and efficiency) of the optmization strongly depends upon the chosen mesh. The authors should discuss this aspect more carefully.
  • No experimental validation is presented. At least in the conclusions, the authors should plan some validation of the numerical results for future studies.
  • For design purposes, syntehtic measures of the optimized shape of metallic dampers should be reported, namely effective stiffness at the target displacement and corresponding equivalent viscous damping ratio, calculated for the various cases shown in Table 1.
  • The review of literature should be improved in the introduction. There is no discussion on the cyclic engagement of hysteretic dampers in seismic retrofit of existing buildings.  This is an important aspect that needs to be discussed. The literature review should be integrated by quoting recent papers in the field, see: “Cyclic engagement of hysteretic steel dampers in braced buildings: A parametric investigation”, Bulletin of Earthquake Engineering 2021; 19:5219-5251.

Author Response

(The authors gave the same response as above.)

Reviewer 3 Report

Comments

This paper investigates the optimization methodology of metallic dampers, with consideration of the intensity of dynamic loads. The two optimization approaches proposed in this study is based on the sequential quadratic programming (SQP) provided by MATLAB and finite element (FE) model established in ABAQUS. The objective of the optimization method is to maximize the energy dissipation of the metallic damper at a given level of seismic excitation. The variables to be optimized is the geometry of the damper. Compared with the massive literatures previously published to address the issue of damper optimization, the novelty of this study is not quite obvious. Therefore, I suggest reconsideration of this paper after major revision. The following are suggestions\comments that may help the authors to improve the quality of the paper:

  1. The introduction part seems very weak. Paragraph 1 focuses on the cons of the magnetoreheological (MR) damper, which is irrelevant to this study. It will be better if the manuscript can start with introduction to metallic damper and studies on its optimization.

  1. Tons of literatures have been published on optimization methods of metallic dampers. The introduction parts can be enriched by discuss all these studies systematically. The gap between the existing method and this study is not fully addressed, damaging the obviousness of its novelty.

  1. Have the authors compared the optimization results with any results reported in other literature?

  1. How is moderate earthquake defined? Is energy dissipation a reasonable objective for the case of moderate earthquake?

  1. Only geometry parameters are optimized in this study. However, in many papers, the topology is considered to find a better shape. Therefore, the novelty of this paper is weak.

  1. There are two approaches proposed in this study, the static approach and the dynamic approach. Have the results been compared? Which method shall be applied for practical engineering?

  1. Since the optimization method relies on the FE model established in ABAQUS, it is essential to assure the accuracy the FE model. Which constitutive model is used to characterized the behavior of the steel elements at the nonlinear stage? Why this model is used?

Minor comments

  1. Page 1 Line 15. What is SQP short for? It shall also be given in both the abstract and the body of the paper.
  2. Page 4. Section 2.2 is missing in the paper. Only Section 2.1 and Section 2.3 exists in the manuscript.
  3. Page 4, Equation (2). E( xi) shall not be enclosed in this equation.
  4. Page 4, Equation (3). εpl is not explained. It is necessary to interpret every term involved in the equations to assist the readers to repeat your study.

Author Response

(The authors gave the same response as above.)

Reviewer 4 Report

This paper provides a very interesting work by optimising the metallic dampers. Generally, this is very good work and deserves to be published with a good contribution to the community. But some technical details are not clear. Some comments can be addressed as follows before a publication is recommended.

[in the abstract]

It is desired to include one sentence to introduce the necessity of developing the metallic damper.

It would be more impactive to see some statistical results in the abstract to show the effectiveness of the present approaches.

[in the introduction]

It is also recommended to indicate the potential application of metallic dampers in some similar engineering backgrounds, such as the transmission lines [1] and overhead contact systems [2].

[1] Wind-Induced Response and Its Controlling of Long-Span Cross-Rope Suspension Transmission Line. Applied Sciences 12.3 (2022): 1488.

[2] Wind deflection analysis of railway catenary under crosswind based on nonlinear finite element model and wind tunnel test." Mechanism and Machine Theory 168 (2022): 104608.

In the literature review, it is recommended to introduce the advantage of metallic dampers against previous ones, after you introduce ‘…many researchers have proposed novel concepts of metallic dampers using…’ for a better logicality.

[in section 2]

A more specific title is recommended other than ‘materials and methods’ to be more informative for readers.

Some outline texts are recommended to insert between the titles of the section and the sub-section to introduce the main content of this section.

It is desired to see more details of the FEM model. How is the model built? with which type of element? is that a two-dimensional or three-dimensional model?

The most interesting point is how to model the metal damper? Some analytical details would be preferred.

[in sections 3&4]

In dynamic optimisation, the excitation to the bridge seems to be quite important to reproduce a realistic response. How do the authors consider this issue? Based on the reviewer’s understanding, only the optimisation based on realistic behaviour has industrial value. Please comment.

Author Response

(The authors gave the same response as above.)

Round 2

Reviewer 1 Report

The authors addressed most comments from my original review. Still, some minor details need to be corrected. Therefore, the authors should solve the following issues before recommending this manuscript for publication.

Points 7, 9, and 17. I suggest you add these responses as comments in your manuscript to state the main assumptions and limitations of your study. You can add comments in the conclusions about the significance of your results given the assumptions mentioned in your response and that future research should tackle these challenges.

Point 11. The unit of volume is not correct. It says [m2], and it should say [m3] or [mm3].

Point 13. Use the unit [Hz] instead of [cycles/sec] across the whole document. Also, it is impossible to obtain stable numerical results in your simulation without damping. I highly suggest inspecting if you add any numerical damping in your simulation and declare this accordingly. Finally, what implicit scheme is used in ABAQUS for time integration? Please clarify.

Point 14. I appreciate you included a discussion on the effect of frequency content in the structural response of the hysteretic damper. Nevertheless, the dynamic response is not complete only with one cycle. Most definitely, the first seconds of response are associated with the transient response, given the initial conditions. The authors should include more time in the simulation to assess also the stationary response. Also, the time when you get a stationary response will depend on the damping ratio, as discussed in point 13 of this review.

Point 16. Add values to the color legend in the stress map in Fig. 13a. Is red color associated with yielding stress? Is Blue zero stress?

Author Response

We appreciate your comments. Please see the attachment.

Reviewer 3 Report

It can be accepted.

Author Response

We appreciate your review.

Reviewer 4 Report

I do not have any further comments. 

Author Response

We appreciate your review.